# Halogen hydrogen-bonded organic framework (XHOF) constructed by singlet open-shell diradical for efficient photoreduction of U(VI)

Lijuan Feng[1], Yihui Yuan [1✉], Bingjie Yan[1], Tiantian Feng[1], Yaping Jian[1], Jiacheng Zhang[1], Wenyan Sun[1], Ke Lin[1], Guangsheng Luo[1] & Ning Wang [1✉]

Synthesis of framework materials possessing specific spatial structures or containing functional ligands has attracted tremendous attention. Herein, a halogen hydrogen-bonded organic framework (XHOF) is fabricated by using $Cl^-$ ions as central connection nodes to connect organic ligands, 7,7,8,8-tetraaminoquinodimethane (TAQ), by forming a $Cl^-\cdots H_3$ hydrogen bond structure. Unlike metallic node-linked MOFs, covalent bond-linked COFs, and intermolecular hydrogen bond-linked HOFs, XHOFs represent a different kind of crystalline framework. The electron-withdrawing effect of $Cl^-$ combined with the electron-rich property of the organic ligand TAQ strengthens the hydrogen bonds and endows XHOF-TAQ with high stability. Due to the production of excited electrons by TAQ under light irradiation, XHOF-TAQ can efficiently catalyze the reduction of soluble U(VI) to insoluble U(IV) with a capacity of 1708 mg-U $g^{-1}$-material. This study fabricates a material for uranium immobilization for the sustainability of the environment and opens up a new direction for synthesizing crystalline framework materials.

[1] State Key Laboratory of Marine Resource Utilization in South China Sea, Hainan University, Haikou 570228, P. R. China. ✉email: yuanyh@hainanu.edu.cn; wangn02@foxmail.com

The design and synthesis of crystalline porous framework materials has garnered numerous attention due to their unique attributes and chemical versatility[1]. There are three main types of crystalline porous frameworks, including covalent organic frameworks (COFs), metal organic frameworks (MOFs), and hydrogen-bonded organic frameworks (HOFs), which are classified according to the types of bonding models[2]. COFs are formed by the connection of covalent bonds and show a stable structure[3–5]. MOFs are formed by the coordination bonds between metallic nodes and organic linkers, which possess high inherent crystallinity but low toughness[6–8]. Hydrogen bond interaction stabilized frameworks include HOFs and supramolecular organic frameworks (SOFs), both of which are composed only of organic moieties and have many unique advantages, including mild synthesis conditions, solvent processability, and easy reconstruction[9,10]. Although substances based on hydrogen bond interactions are known, they have not received much attention until recently due to their potential applications as functional frameworks[11,12]. Current studies of hydrogen bond-based crystalline porous frameworks are mainly focused on N···H or O···H hydrogen bonds. However, the halogen group, as the most electronegative element in the same period of the periodic table, is more likely to form hydrogen bonds and work as a connection node for the framework material[13,14]. Generally, halogen bonds can be described as D···X—Y, in which X represents the halogen atom, D represents the hydrogen atom, and Y is a highly electronegative atom, such as N, O, and F, that play a key role in the control of intermolecular recognition and self-assembly[15,16]. D···X—Y are used to construct unique supramolecular architectures that have been widely used in crystal fields, such as molecular recognition, catalysis, and drug design[17–21]. However, the D···X···D-type halogen-based hydrogen-bonded organic framework is still rarely reported. Compared with the D···X—Y bond, D···X···D shows the advantages of being easy to self-assemble, convenient to functionalize, and highly tunable. Thus, developing new connection types based on D···X···D is important for expanding the family of crystalline porous frameworks with undeveloped features and functions (Fig. 1a).

In addition, organic building blocks also affect the properties of the framework materials. The organic building blocks not only play the role of framework construction but also provide diverse properties, including optical and electrical properties[22,23]. By seeking suitable building blocks and functional organic groups, crystalline porous framework materials can exhibit a strong visible light absorption ability and fast charge carrier mobility[24,25]. Therefore, it is of great significance to develop new functional organic ligands for the construction of crystalline porous frameworks. In 1907, Chichibabin first reported a quinoid structure, p,p′-biphenylene-bis-(diphenylmethyl), whose resonance structure is the singlet open-shell diradical form (Fig. 1b)[26,27]. Compared with aromatic hydrocarbons, quinone hydrocarbons exhibit extra double-bond properties outside the aromatic ring, which is beneficial for electron delocalization[28]. Thus, these materials can be used as organic semiconductor materials[29–31]. Additionally, due to their unique electronic structures, singlet open-shell diradicals exhibit interesting properties in photonics, electricity and magnetism[32,33]. As a consequence, with the rejuvenation of free radical chemistry, singlet open-shell diradicals have become a research hot spot, and their applications in photocatalytic reactions have also been explored[34,35]. However, the instability of diradicals hinders their development[36]. Therefore, the design and synthesis of stable singlet open-shell diradicals to construct organic frameworks remains of significance and a great challenge.

Uranium, an important strategic resource, has limited reserves[37–39]. The contamination of uranium in nuclear waste liquids is harmful to human health and the ecological environment due to its chemical toxicity and radiotoxicity[40]. Thus, efficient removal of uranium contamination and recovery of uranium resources via a secondary mineralization strategy are both of great significance to the sustainable development of the nuclear industry and the environment[41–46]. In this work, based on the 7,7,8,8-tetraaminoquinodimethane (TAQ) ligand and chloride ion ($Cl^-$), a crystalline framework material, designed as a halogen hydrogen-bonded organic framework (XHOF), is constructed. Structural analysis shows that $Cl^-$ works as a connection node to connect three adjacent ligands through three hydrogen bonds to form a regular 3D structure framework, which represents a different kind of framework material (Fig. 1c). Compared with MOFs, the halogen atom in XHOF replaces the metal atom to work as a connection node[47]. Although the HOF also uses hydrogen bonds to construct the framework, hydrogen bonds are formed directly between the ligands, while the XHOF utilizes central halogen atoms to connect the ligands. The light-induced singlet open-shell diradical structure of TAQ provides excited electrons to efficiently catalyze the reduction of highly soluble U(VI) to insoluble U(IV), which endows XHOF-TAQ with high potential for immobilizing uranium (Fig. 1d).

## Results

### Fabrication of XHOF-TAQ and characterization of the organic ligand

Terephthalamide oxime (TPAO), the main raw material used for the synthesis of XHOF-TAQ, was prepared by oximation of terephthalonitrile (TPN) (Fig. 2a). The Fourier transform infrared (FTIR) spectra for TPAO and TPN show that the nitrile group of TPN is successfully oximated into amidoxime group in TPAO, represented by the disappearance of the $C\equiv N$ peak ($2230\ cm^{-1}$) (Supplementary Fig. 1). The results from mass spectrum (MS) analysis show that the main peak appears at 195, which is responsible for TPAO, whose molecular weight is 194 Da (Fig. 2b). Subsequently, the synthesized TPAO and $CuCl_2\cdot 2H_2O$ were used to synthesize XHOF-TAQ by the solvothermal method, and colorless octahedral crystals were obtained. Based on the usage of $Cl^-$, the yield of the crystals was calculated to be 46%. The analysis of the chemical components of the crystal shows that the C, H, N, and Cl contents are 40.53%, 5.14%, 23.59%, and 29.70%, respectively, which match the contents in the chemical formula $C_4H_6N_2Cl$. Interestingly, the raw material TPAO contains O, while there is no O in the crystal based on chemical component analysis. Energy-dispersive X-ray spectroscopy (EDS) elemental mapping analysis coupled with scanning electron microscopy (SEM) confirmed the presence of Cl and N and the absence of O in the crystal (Fig. 2c). This result indicates that the structure of the organic ligand TPAO is changed during the synthesis process. Based on the chemical component of the crystal, compared with the structure of TAPO, the organic ligand in the crystal structure is found to lose two hydroxyl groups (OH) and gain four hydrogen atoms, which is consistent with the structure of TAQ. The MS analysis for the organic ligand in the crystal shows that one peak with a molecular weight of 163 Da is detected, which corresponds to TAQ (Supplementary Fig. 2). The $^1H$ NMR spectrum of XHOF-TAQ shows that the hydrogen atoms on the amino is twice the benzene ring, which further confirms the structure of the TAQ ligand (Fig. 2d). These results all prove that TAQ instead of TPAO participates in the synthesis of XHOF-TAQ. The loss of the hydroxyl group from the oxime group is due to the catalysis of $Cu^{2+}$ during the solvothermal synthesis process[48,49]. Other $MCl_n$ compounds (M = Mn, Zn, Co, Ni, etc.) have also been used to replace $CuCl_2$ in the synthesis process, but no observable crystal product has been synthesized, indicating

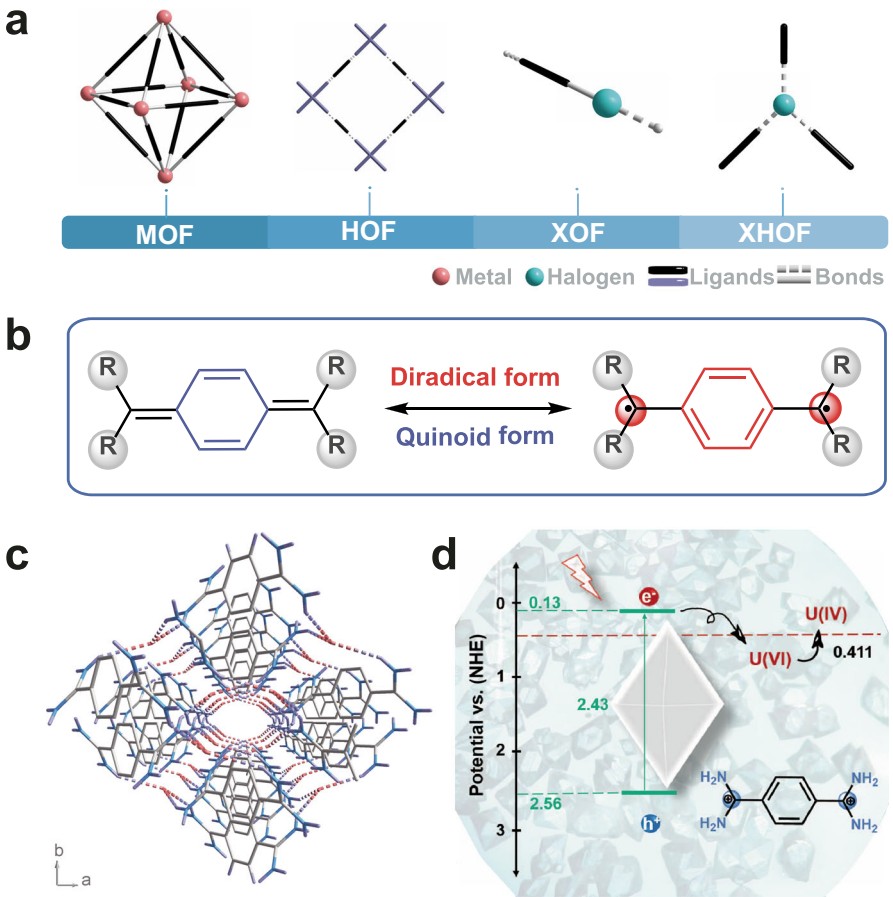

**Fig. 1 Schematic diagram of the mechanism for U(VI) immobilization by XHOF-TAQ. a** Different connection models of the organic framework materials. **b** Resonance structures of the quinoid and diradical forms. **c** Illustration of the 3D structure of XHOF-TAQ. **d** Mechanism for electron and hole transfer during the photoreduction of U(VI).

that Cu$^{2+}$ is essential for the synthesis of the crystalline framework material for its catalytic activity. Compared with TPAO, TAQ is not stable and easily oxidized in air[26]. The use of TPAO as the initial substrate does not entail control of the oxygen concentration for the reaction and can reduce the reaction conditions for the synthesis of XHOF-TAQ. Furthermore, the replacement of CuCl$_2$ with other CuX$_2$ compounds (X = F, Br, and I) also leads to failure in the synthesis of crystal products because the other halogen ions cannot take part in the synthesis of such framework materials or the reaction conditions are not suitable for the synthesis of such framework materials containing other halogen ions.

**Crystal structure and characterization of XHOF-TAQ.** The fine structure of XHOF-TAQ was determined by single-crystal X-ray diffraction (SC-XRD) analysis. The crystal structure of XHOF-TAQ exists as a C2/c space group in the monoclinic crystallized system. Each asymmetric unit of XHOF-TAQ contains two Cl$^−$ ions and one TAQ ligand. Although Cl shows a weaker tendency to form hydrogen bonds than F, benefiting from the abundant amino groups working as electron donors around the chloride ion, one Cl$^−$ forms three hydrogen bonds with three amino hydrogen atoms from three TAQ ligands (Fig. 3a). These Cl$^−$···H$_3$ hydrogen bonds form a stable approximate plane triangle structure and endow the crystal with high stability. The lengths of the three hydrogen bonds between Cl$^−$ and the H atom are 2.33 Å, 2.39 Å, and 2.44 Å, respectively. The planes are spatially independent of each other, and the nearest distance between the central Cl$^−$ of the plane is 4.36 Å, which indicates that there is no interaction between the nearest Cl$^−$, and, thus, more

halogen hydrogen bonds can be formed between Cl$^−$ and the organic ligands to provide XHOF-TAQ with higher stability[15]. Finally, the planes are further connected by TAQ ligands to form a 3D structure. The structure has been deposited at the Cambridge Crystallographic Data Centre (CCDC) under number 2096137. The X-ray photoelectron spectra (XPS) analysis for CuCl$_2$ and XHOF-TAQ reveal that the binding energy of Cl$^−$ shifts simultaneously from 200.40 eV and 198.82 eV in CuCl$_2$ to 198.82 eV and 197.23 eV in XHOF-TAQ, indicating that the outer layer electron density of Cl$^−$ is increased (Fig. 3b and Supplementary Fig. 3). The increase in the electron density of Cl$^−$ is attributed to the sharing of electrons from the electron-rich organic TAQ ligand to maintain a localized negatively charged center[50]. The electron-rich characteristic of TAQ is the reason why it is easily oxidized in air. This kind of electron sharing reduces the electron-rich property and increases the air stability of TAQ in XHOF-TAQ.

The FTIR analysis of XHOF-TAQ shows that the major peaks are consistent with the chemical groups present on the organic ligand in the crystal (Supplementary Fig. 4). Powder X-ray diffraction (PXRD) was used to identify the XHOF-TAQ phase, and the PXRD spectrum of synthetic XHOF-TAQ is found to fit well with the simulation data, indicating that the XHOF-TAQ phase is pure (Fig. 3c). The PXRD patterns for XHOF-TAQ treated with different solvents show that no framework collapse or phase transition occurs after contact with solvents for 6 h, suggesting that XHOF-TAQ is highly stable under diverse solvents (Supplementary Fig. 5). XHOF-TAQ also shows high environmental stability. After being placed in air for one year, the material still maintains its initial crystal structure (Supplementary Fig. 6). A thermogravimetric analyzer (TGA) was used to test the

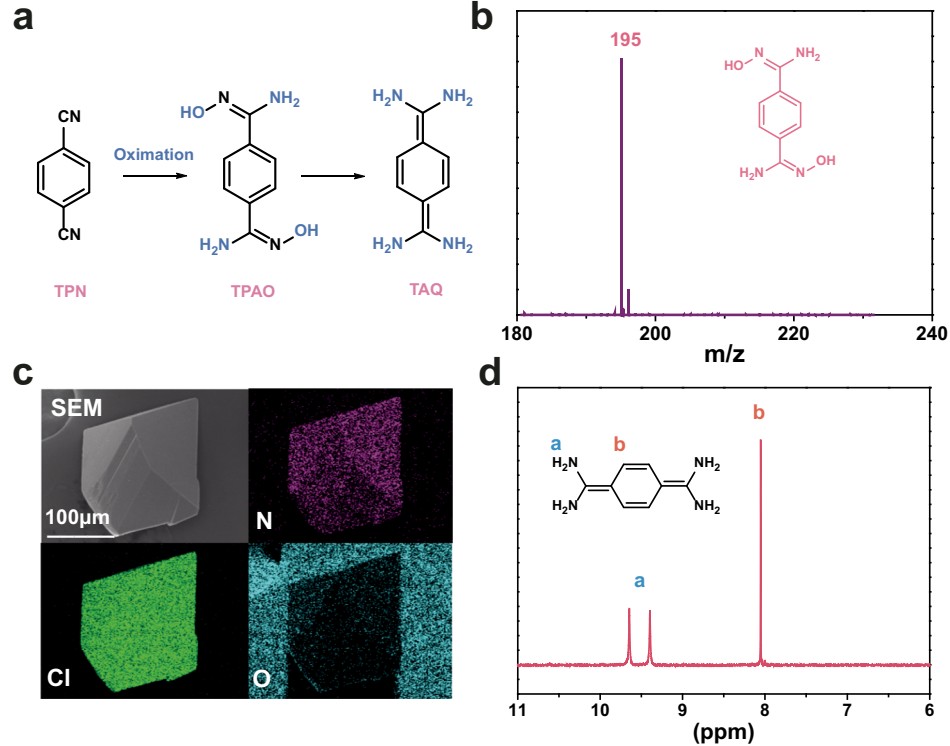

**Fig. 2 Synthesis of XHOF-TAQ and characterization of the organic ligand. a** Schematic diagram showing the changes of ligands for fabricating XHOF-TAQ. **b** The MS spectrum of TPAO. **c** SEM image and EDS mapping of XHOF-TAQ. **d** $^{1}$H NMR spectrum of XHOF-TAQ.

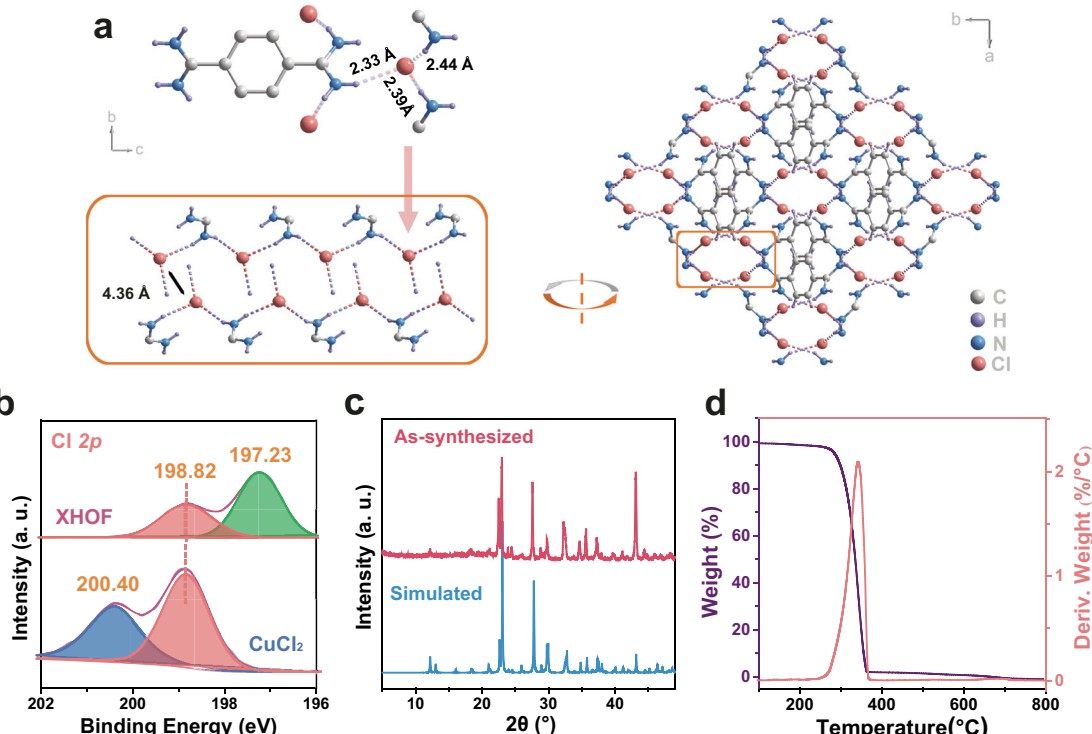

**Fig. 3 Crystal structure and characterization of the materials. a** Crystal structure of XHOF-TAQ. **b** High-resolution XPS for Cl. **c** The simulated and as-synthesized PXRD patterns for XHOF-TAQ. **d** TGA analysis of XHOF-TAQ.

thermal stability of XHOF-TAQ. The result shows that no significant weight loss occurs before 280 °C, and continuous weight loss occurs in the range of 280 °C to 363 °C, which is attributed to the decomposition of the TAQ ligand (Fig. 3d). Finally, almost all the weight is lost above 363 °C. The TGA analysis indicates that the hydrogen bonds in XHOF-TAQ can also endow XHOF-TAQ with good thermal stability, which is comparable to other framework materials[51,52].

**Determination of the photoreduction mechanism for XHOF-TAQ to U(VI).** Conversion of soluble hexavalent uranium (U(VI)) to insoluble tetravalent uranium (U(IV)) is an efficient method for uranium immobilization[53–56]. In consideration of the existence of the photoinduced diradical form of TAQ, XHOF-TAQ is highly likely to possess a photoreduction ability for immobilizing uranium. Thus, the photoreduction ability of XHOF-TAQ to U(VI) was determined. The result shows that with a dosage of 0.05 g L$^{-1}$, XHOF-TAQ can immobilize U(VI) in a solution with a high immobilization capacity of 1708 mg-U g$^{-1}$-material. Compared with the other materials available for the photoreduction of uranium, XHOF-TAQ exhibits a high uranium immobilization capacity together with a fast uranium immobilization speed of 34.16 mg g$^{-1}$ min$^{-1}$ under light irradiation (Supplementary Table 1). The change in the forms of uranium was determined by detailed high-resolution XPS analysis. The result shows that before the reduction, the characteristic peaks for U4f 5/2 and U4f 7/2 appear at 392.68 eV and 381.87 eV, respectively (Fig. 4a and Supplementary Fig. 7). After the treatment of XHOF-TAQ and light irradiation, new peaks for U4f 5/2 and U4f 7/2 of U(IV) appear at 392.01 eV and 381.16 eV, respectively, indicating that U(VI) is reduced to U(IV). The PXRD analysis for the used XHOF-TAQ shows that the used material still maintains its initial crystal structure, which proves the stability of the material and the feasibility of the material for practical application (Supplementary Fig. 8).

The photoreduction mechanism for XHOF-TAQ to U(VI) was investigated by experimental analysis together with density functional theory (DFT) calculations. Electron paramagnetic resonance (EPR) measurements of XHOF-TAQ show that a featureless wide signal ($g_e = 2.00$) is observed only in the presence of light (Fig. 4b). The variable temperature EPR tests show that the intensity increases with increasing temperature in the presence of light (Supplementary Fig. 9). The fitting of the variable temperature EPR intensities by the Bleaney-Bowers equation shows that the singlet-triplet energy gap ($\Delta E_{s-t}$) is −1.22 kcal mol$^{-1}$, which confirms the singlet diradical state for XHOF-TAQ under light irradiation (Supplementary Fig. 10)[57]. To further elucidate the electronic structure of XHOF-TAQ and describe the exchange-correlation energies, DFT calculations were performed using the Gaussian 09 package with the B3LYP (Becke, three-parameter, Lee-Yang-Parr) hybrid function. The energy of the TAQ singlet open-shell diradical state is lower than that of the closed-shell state by 1.21 kcal mol$^{-1}$, which proves the higher potential of TAQ to form singlet open-shell diradical ground electronic states. The electron spin density distribution of TAQ was also calculated. The calculated singly occupied molecular orbital (SOMO) for the α and β spin profiles shows a disjoint character (Fig. 4c). The analysis of the spin density distribution of TAQ shows that the carbon atoms in the ring have smaller amplitudes, while the C7 and C8 atoms have much larger amplitudes, suggesting that the diradical appears at C7 and C8 atoms (Fig. 4d). Based on all of the above findings, TAQ can be described as having a resonant structure between quinoid and biradical form.

Valence band XPS and UV-visible diffuse reflectance spectroscopy (DRS) for XHOF-TAQ were carried out to further confirm the electron excitation process. The valence band XPS spectrum shows that the edge of the valence band (Ev) of XHOF-TAQ is 2.56 eV (Fig. 4e). DRS analysis shows that the absorption edge of XHOF-TAQ is 450 nm (Supplementary Fig. 11). The DRS spectrum was converted to a TAUC plot by using the equation, $\alpha(h\nu) = (\alpha h \nu)^2$, where α, h, and ν are the absorption coefficient, Plank's constant, and light frequency, respectively. Then, by extrapolating the tangent line to the X-coordinate, a positive slope is observed, illustrating that XHOF-TAQ is an *n*-type

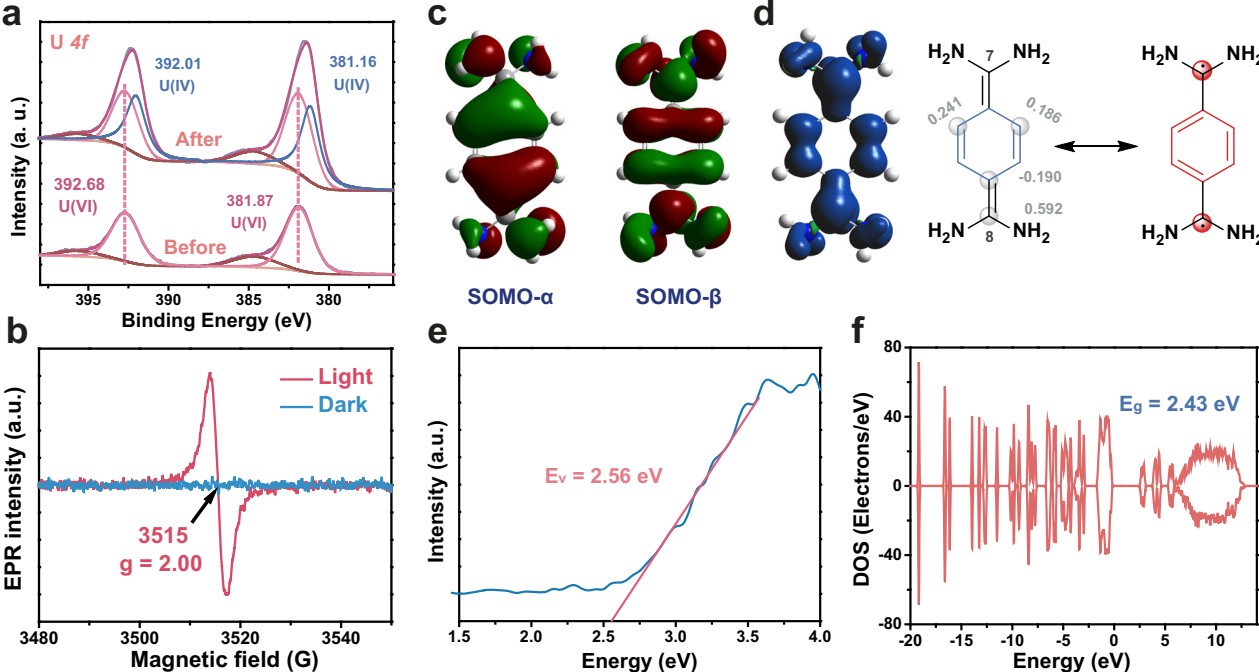

**Fig. 4 Mechanism for the photoreduction of U(VI). a** High-resolution XPS spectra of U before and after photoreduction. **b** EPR spectra of XHOF-TAQ in the dark and under light irradiation. **c** Calculated SOMOs for the α and β spin profiles of XHOF-TAQ. **d** Calculated spin density distribution of XHOF-TAQ. **e** The valence band XPS for XHOF-TAQ. **f** Calculated density of electronic states for XHOF-TAQ.

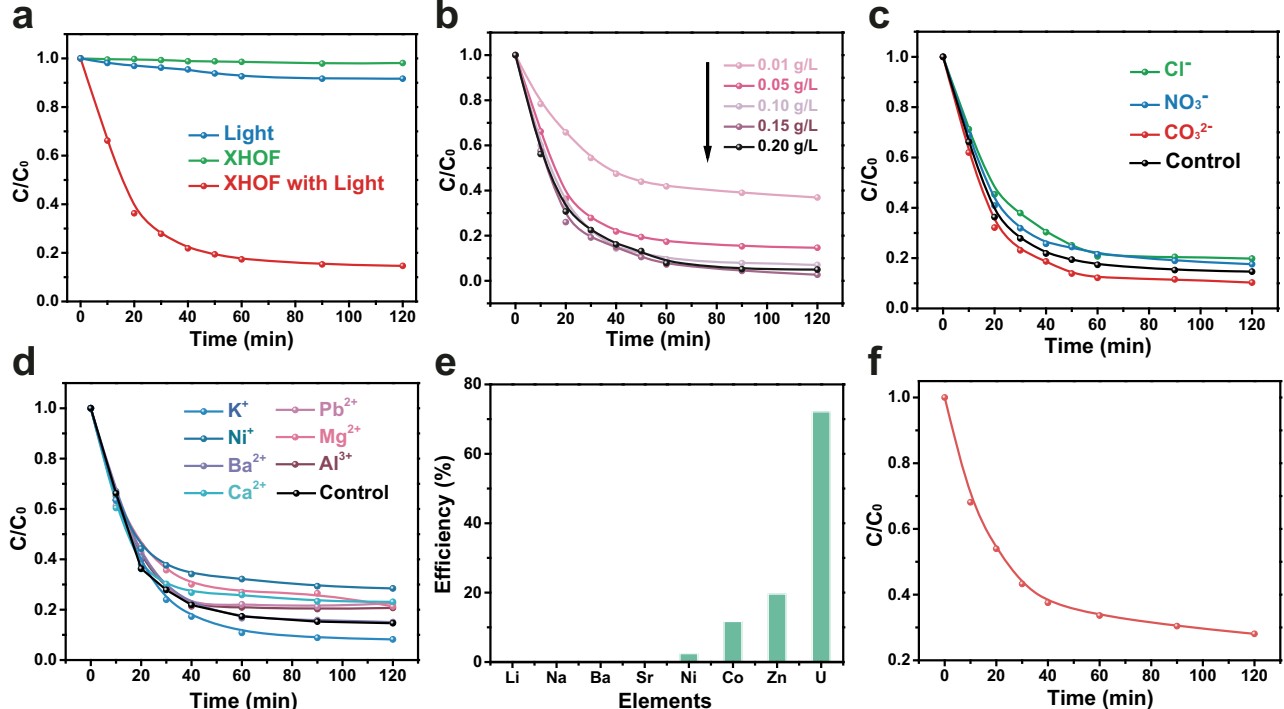

**Fig. 5 The photoreduction performance of XHOF-TAQ to U(VI). a** Essential factors for the photoreduction of U(VI) by XHOF-TAQ. **b** Influence of XHOF-TAQ dosage on the photoreduction efficiency. **c** Effect of anions on the photoreduction efficiency. The sodium salt of each anion with a concentration of 10 mM is used. **d** Effect of cations on the photoreduction efficiency. The chloride salt of each cation with a concentration of 10 mM is used. **e** Effect of mixed interfering metal ions on the photoreduction efficiency. The nitrate of each metal ion with an individual concentration of 10 mM was used. **f** The photoreduction kinetics obtained for U in co-existence with mixed interfering metal ions.

semiconductor. Based on the TAUC plot, the band gap (Eg) of XHOF-TAQ is estimated to be 2.43 eV, which is consistent with the Eg value obtained from the DFT calculations for the density of electronic states (Fig. 4f and Supplementary Fig. 12)[58]. Accordingly, the conduction band (Ec) is calculated to be 0.13 eV, confirming the photoreduction ability of XHOF-TAQ to U(VI). Under light irradiation, TAQ in XHOF-TAQ forms a diradical structure, which produces excited electrons to catalyze the reduction of U(VI) to U(IV).

**Photocatalytic immobilization of U(VI).** The efficiency of photocatalytic immobilization of U(VI) by using XHOF-TAQ in the posttreatment of nuclear waste liquid was analyzed (Fig. 5a). Neither irradiation with light alone nor XHOF-TAQ treatment alone leads to a significant decrease in the concentration of U(VI), indicating that both XHOF-TAQ and light irradiation are essential for the immobilization of U(VI) and that XHOF-TAQ does not possess a uranium adsorption ability. The photoreduction performance was optimized by adjusting the XHOF-TAQ dosage from 0.01 to 0.20 g L$^{-1}$ at room temperature without removing oxygen (Fig. 5b). The results show that the photoreduction efficiency exhibits dosage-dependent characteristics. For an increase in the dosage from 0.01 to 0.10 g L$^{-1}$, the catalytic efficiency is rapidly increased, while a further increase in the dosage only leads to a small improvement in the photoreduction efficiency. Approximately 92.0% immobilization efficiency is reached within 90 min under light irradiation with a catalyst dosage of 0.10 g L$^{-1}$, whose immobilization capacity is calculated to be 921.7 mg-U g$^{-1}$-material, indicating that XHOF-TAQ maintains a high photoreduction ability for U(VI) under light irradiation.

Coexisting cations and anions have been reported to influence the speciation of uranium or affect electron transport by binding to the surface of the material[42]. The influence of environmental cations and anions on the photoreduction efficiency was analyzed to determine the environmental adaptability of XHOF-TAQ. For the anions, the coexistence of $CO_3^{2-}$ can boost the photoreduction efficiency, while $Cl^-$ and $NO_3^-$ only slightly influence the photoreduction efficiency (Fig. 5c). For the cations, $K^+$ boosts the photoreduction efficiency, and $Ba^{2+}$ shows no interference with the photoreduction efficiency (Fig. 5d). The enhancement effect of $K^+$ on the photoreduction efficiency is attributed to the large ion radii, which provide a large intermolecular space for electron transport[59]. With an approximately 50 times higher molar concentration for the other interfering cations to U(VI), XHOF-TAQ still retains a photoreduction efficiency of more than 71.6%. Cations such as $Ni^{2+}$, $Co^{2+}$, and $Zn^{2+}$ can interfere with the photoreduction ability by occupying the excited electrons, which is proven by the immobilization of these metals (Fig. 5e). However, from a calculation of the immobilization capacity, the photoreduction ability for uranium is at least 3.68 times higher than that for the other metals, proving the high selectivity of XHOF-TAQ in the photoreduction of uranium. Even in the solution containing multi-metal ions, which all exist with an approximately 50 times higher molar concentration, XHOF-TAQ still shows a photoreduction efficiency of 72.0% to U(VI) within 120 min (Fig. 5f).

## Discussion

In this study, a crystalline framework material, XHOF, is synthesized by using halogen ions as connecting nodes to form multihydrogen bonds for the architecture of the framework, which is different from the connection models of MOF, COF, and HOF and represents an unexplored kind of framework material. The electron-withdrawing effect of $Cl^-$ and the electron-rich property of TAQ strengthens the hydrogen bonds for the architecture of the crystal structure and endows XHOF-TAQ with high stability in different solvents. In addition, the TAQ ligand in XHOF-TAQ possesses a singlet open-shell diradical structure under light irradiation, which can release excited electrons to

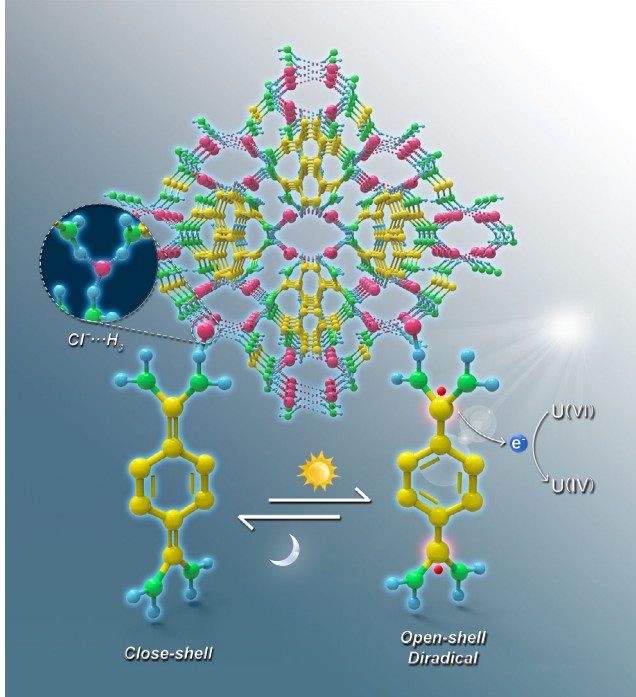

**Fig. 6 Schematic diagram of the mechanism for photoreduction of U(VI) by XHOF-TAQ.** The connection model for constructing the framework material is also shown.

catalyze the reduction of soluble U(VI) to insoluble U(IV) for immobilization of uranium from the liquid environment (Fig. 6). With a low dosage of $0.10 \, g \, L^{-1}$, XHOF-TAQ exhibits a high reducibility of 93.0%. Furthermore, due to its suitable band gap and band position, XHOF-TAQ shows a selective photoreduction ability for uranium. In summary, this study synthesizes an XHOF-TAQ material based on connections formed by $Cl^{-}\cdots H_3$ hydrogen bonds. Moreover, this is also the first report for the introduction of chemically unstable singlet open shell diradicals into framework materials by chemical reactions during a solvothermal synthesis process. The findings of this study not only provide a strategy for the immobilization of uranium but also open up a new direction for the synthesis of framework materials for potential applications.

## Methods

**Materials and characterization**. TPAO was synthesized by the amidoximation of terephthalonitrile (TPN) in organic solvent, as described below. All other chemicals were obtained from commercial Macklin and used without further purification. XRD was carried out by using a Haoyuan DX-2700BH X-ray diffractometer, and the simulated powder pattern was output using Mercury software. The FTIR spectrum (KBr pellets) was recorded using a PerkinElmer Frontier FT-NIR/MIR spectrometer at room temperature. TG-DTA analysis was performed using a Netzsch STA 449 F5 Jupiter thermal analyzer from room temperature to 800 °C with a heating rate of 10 °C $min^{-1}$ under a $N_2$ atmosphere. The XPS spectra were obtained using a Kratos AXIS SUPRA spectrometer. ICP-MS was conducted using an Agilent ICPMS7899 instrument. The DRS spectra were measured using a Shimadzu UV-3600 spectrophotometer, and $BaSO_4$ was used as a reflectance standard. $^{1}$H NMR spectra were acquired using a Brüker Advance 400, with DMSO-D6 and TMS used as the solvent and internal standard, respectively. The MS data were recorded using a Bruker ultrafleXtreme MALDI-TOF analyzer. The ESR data were measured using a Bruker A300-12 spectrometer.

**Preparation of TPAO**. $NH_2OH\cdot HCl$ (24.00 g, 0.34 mol) was first completely dissolved in DMF (100 mL) solution, and then NaOH (6.00 g, 0.15 mol) and $Na_2CO_3$ (7.80 g, 0.74 mol) were added to the solution at 45 °C. After vigorous stirring for 2 h to neutralize the solution, the terephthalonitrile (TPN, 22.00 g, 0.17 mol) powder was added to the mixture. After stirring continuously for 16 h at

75 °C, the resulting mixture was centrifuged to separate the undissolved particles, and then the supernatant was collected for use.

**Fabrication of XHOF-TAQ**. The synthesized TPAO (0.4 mL, 0.68 mmol) together with $CuCl_2\cdot 2H_2O$ (0.17 g, 1.00 mmol) was added to DMF (5 mL), and then the mixture was treated with ultrasonication at room temperature for 10 min After that, the reaction solution was sealed in a 10 mL vial and heated at 95 °C for 5 days. After cooling to room temperature, the reaction mixture was isolated by filtration, washed with DMF, and then dried in air. Based on the usage of $Cl^{-}$, the yield of the product was calculated to be 46%. 1H NMR (400 MHz, DMSO): δ 8. 04 (s, 1H), 9.39 (s, 1H), 9.64 (s, 1H); IR (KBr pellets): 3214 (s), 3036 (s), 1695(s), 1656(s), 1544(s), 1473(s), 879 (m), and 698(s); element analysis (calcd., found for $C_4H_6N_2Cl$): C (40.86, 40.53), H (5.14, 5.14), Cl (30.16, 29.70), N (23.84, 23.59).

**X-ray crystallography**. The diffraction data were measured using Bruker D8 Venture equipment at 150 K with Cu Kα radiation (λ = 1.54178 Å). The crystal structure was solved using Olex2 software with the XS structure solution program with direct methods, and then further refined with the XL refinement package using least-squares minimization. All the nonhydrogen atoms were refined with anisotropic thermal displacement coefficients. The hydrogen atoms were included in idealized positions and refined by a riding model. The data details for XHOF-TAQ are listed in Supplementary Table 2.

**Computational method**. All DFT calculations were performed by using Gaussian 09 software in the framework. All the molecular structures were computed with the Becke three-parameter Lee-Yang-Parr (B3LYP) hybrid functional method to describe the exchange-correlation energies. The Los Alamos LANL2DZ effective core pseudopotentials (ECPs) were applied to Cl. The atoms in the ligands, including C, N, and H, were calculated using high-level B3LYP calculations with 6-31G+(d,p) basis sets. To confirm its identity as an energy minimum, vibrational analysis was performed at each stationary point, and the natural bond orbital method was also performed for the population analysis.

**Photocatalysis experiments**. For the photocatalysis experiment, a 300 W Xe lamp (200–800 nm) with a light density of 1 kW $m^{-2}$ was used as the source of simulated sunlight irradiation at room temperature. The photocatalyst XHOF-TAQ was added to 20 mL of 50 ppm uranium in DMF without sacrificial agents. The suspension was stirred in the dark for 20 min to achieve reaction equilibrium. During the photocatalysis process, 100 μL suspension was removed and filtered with a nylon syringe membrane at regular intervals for monitoring the uranyl concentration. The concentration of U(VI) in the filtrate was determined by ICP-MS.

## Data availability

The data supporting the findings of this study are available in the paper and its Supplementary Information. Source data are provided along with this paper. The X-ray crystallographic coordinates for the structure reported in this study have been deposited in the Cambridge Crystallographic Data Centre (CCDC), under deposition number CCDC-2096137. Source data are provided with this paper.

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

## Acknowledgements

This work is supported by the Hainan Science and Technology Major Project (ZDKJ2020011 awarded to NW and ZDKJ2019013 awarded to YY), the National Natural Science Foundations of China (No. 41966009 awarded to YY, U1967213 awarded to NW, U2167220 awarded to YY), and the National Key R&D program of China (2018YFE0103500 awarded to NW).

## Author contributions

N.W., Y.Y., & L.F. conceived the research and designed the experiments. L.F., B.Y., T.F., Y.J., J.Z., W.S., K.L., & G.L. carried out the experiments. All authors analyzed the data. L.F., Y.Y., & N.W. contributed to the project discussions. L.F., Y.Y., & N.W. wrote the paper.

## Competing interests

The authors declare no competing interests.
