## [Peer Review File · Nature Communications]

REVIEWER COMMENTS

Reviewer #1 (Remarks to the Author):

Synthesis of novel framework materials gains tremendous attentions in recent years. This manuscript reports a new kind of crystalline framework material with high uranium reduction ability. This kind of framework material shows novel construction mode, which is different from the existing MOF, COF, and HOF. The halogen atom is found to act as connection node for the first time. The ligand used for construction of the framework material is also new. What's important is that this material shows high uranium reduction ability and can be used in the recovery of uranium resources. This is a new finding and an important progress in the relate area. The manuscript is well organized and the result is well presented. I would like to recommend the publication of this manuscript in Nature Communications after some minor revision.

1.As described in the manuscript, XHOF represents a new kind of framework materials. The detailed differences between XHOF and the other framework materials should be emphasized more to show the novelty of this study, which have not been well emphasized.

2.For synthesis of XHOF-TAQ, TPAO is used as initial substrate, but only TAQ is detected in the synthesized framework material due to the happen of potential chemical reaction. This is an interesting finding. The author state that "the use of TPAO as the initial substrate can reduce the reaction condition for the synthesis of XHOF-TAQ". Why the use of TPAO can reduce the reaction condition for material synthesis? More descriptions about this statement should be provided to make this sentence clearly.

3.The author mentions that singlet open-shell diradical is not stable and easily to be oxidized in air. What's the force to stabilize TAQ in XHOF?

4.How about the stability of the material after being used for uranium reduction? More characterization should be provided to show the feasibility of this material for practical application.

5."The DRS results is consistent with the E_g value obtained from the DFT calculations of the density of electronic states." How does the E_g value obtained from DFT calculate in the Fig. 4f? Some more references should be cited in this section.

6.The figure in the sentence "...which produce excited electrons to catalyze the reduction of U(VI) to U(IV) (Fig. 1c)" is wrongly written. Fig. 1c should be corrected to Fig. 1d.

Reviewer #2 (Remarks to the Author):

This study reports the fabrication of a novel kind of framework material and the potential application of this material in photoreduction of uranium. This material shows a high uranium recovery capacity and the mechanism for uranium reduction has also been uncovered. This study is an important progress in the area of new crystalline framework materials and uranium

immobilization. The finding of this study is interesting to researchers of a broad area. The manuscript is well organized and the result of this study is well presented. The manuscript can be accepted for publication in Nature Communication after addressing the following minor concerns.

1. The Cl⁻ is reported to work as connection nodes in the material fabricated in this study. Can the other halogen ions, such as F⁻, Br⁻, I⁻, can also work as connection nodes for the construction of this kind of material?
2. As the major ligand in the material, TAQ is not stable in air. How about the environmental stability of the material in air?
3. Cu²⁺ is stated to be essential for synthesis of the material. What's the reason for the essential of Cu²⁺ in the synthesis of this kind of framework material?
4. What's the purpose to provide the nearest distance between the central Cl⁻ in the manuscript? Dose such characterization associate with the structure stability or the uranium reduction ability of the material?
5. The author states that the material fabricated in this study shows comparable thermal stability to other framework materials. Extra references should be provided to support this conclusion.
6. How about the photoreduction selectivity of the material? Can it cause the photoreduction of the other metal ions?
7. How about the uranium photoreduction performance of this material comparing with the other available materials? Including the efficiency and the speed of material in uranium reduction.

Title: Halogen hydrogen-bonded organic framework (XHOF) constructed by singlet open-shell diradical for efficient photoreduction of U(VI)

Response to Reviewers' Comments

Reviewer #1:

General Comments:

Synthesis of novel framework materials gains tremendous attentions in recent years. This manuscript reports a new kind of crystalline framework material with high uranium reduction ability. This kind of framework material shows novel construction mode, which is different from the existing MOF, COF, and HOF. The halogen atom is found to act as connection node for the first time. The ligand used for construction of the framework material is also new. What's important is that this material shows high uranium reduction ability and can be used in the recovery of uranium resources. This is a new finding and an important progress in the relate area. The manuscript is well organized and the result is well presented. I would like to recommend the publication of this manuscript in Nature Communications after some minor revision.

General response: We thank the reviewer for taking time to review our manuscript and the positive/valuable comments and suggestions to improve the manuscript. We have revised the manuscript carefully and added extra experimental data in accordance with the comments. All the revisions in the manuscript have been highlighted in blue. We hope that the revised manuscript is acceptable.

1. As described in the manuscript, XHOF represents a new kind of framework materials. The detailed differences between XHOF and the other framework materials should be emphasized more to show the novelty of this study, which have not been well emphasized.

Response: We thank the reviewer for the valuable comments. The detailed differences between XHOF and the other framework materials, including MOF and HOF, which show the highest similarity, have been added into the revised Introduction section to emphasize the novelty of this study. Comparing with MOFs, the halogen atom in

XHOF replaces the metal atom to work as connection nodes. Although HOF also use hydrogen bonds to construct framework, the hydrogen bonds are forming directly between the ligands, while XHOF uses central halogen atoms to connect the ligands.

Please refer to **Page 5** in the revised manuscript:

“The structure analysis shows that Cl^- works as connection nodes to connect three adjacent ligands through three hydrogen bonds to form a regular 3D structure framework, which represents a novel kind of framework materials (Fig. 1c). Comparing with MOFs, the halogen atom in XHOF replaces the metal atom to work as connection nodes.⁴⁷ Although HOF also use hydrogen bonds to construct framework, the hydrogen bonds are forming directly between the ligands, while XHOF uses central halogen atoms to connect the ligands.”

Please refer to Reference 47 in the revised manuscript:

“Gilday, L. C. et al. Halogen Bonding in Supramolecular Chemistry. *Chem Rev* 115, 7118-7195 (2015).”

2. For synthesis of XHOF-TAQ, TPAO is used as initial substrate, but only TAQ is detected in the synthesized framework material due to the happen of potential chemical reaction. This is an interesting finding. The author state that “the use of TPAO as the initial substrate can reduce the reaction condition for the synthesis of XHOF-TAQ”. Why the use of TPAO can reduce the reaction condition for material synthesis? More descriptions about this statement should be provided to make this sentence clearly.

Response: We thank the reviewer for the valuable comments. According to previous report, cooper can catalyze the transformation of TPAO to form TAQ (*Adv. Synth. Catal.* **2018**, 360, 334-345). In this study, the CuCl_2 is used for preparing the material. Thus, the TPAO is transformed into TAQ after the fabrication of the materials. Compared with TAQ, TPAO is stable in air. Thus, the use of TPAO doesn't have to control the reaction condition, such as the oxygen concentration, for synthesis of the material. We have added additional statements to make the sentence clearly in the revised manuscript.

Please refer to **Page 8** in the revised manuscript:

“The use of TPAO as the initial substrate doesn’t have to control the oxygen concentration for the reaction and can reduce the reaction condition for the synthesis of XHOF-TAQ.”

3. The author mentions that singlet open-shell diradical is not stable and easily to be oxidized in air. What’s the force to stabilize TAQ in XHOF?

Response: We thank the reviewer for the valuable comments. The unstable property of the singlet open-shell diradical form of TAQ in air is attributing to its’ electron rich character, which is easily to be oxidized by the oxygen. After the formation of XHOF, the electron of TAQ is sharing with the Cl^- through forming hydrogen bonds and the electron rich character of TAQ is reduced, resulting in the increase of the stability of TAQ in XHOF. We have added the statements about the reason in the revised manuscript.

Please refer to **Page 10** in the revised manuscript:

“The increase of the electron density of Cl^- is attributed to the share of electrons from the electron rich organic ligand TAQ to maintain a localized negatively charged center.⁵⁰ The electron rich characteristic of TAQ is the reason for it is easy to be oxidized in air. Such kind of electron sharing reduce the electron rich property and increase the air stability of TAQ in XHOF-TAQ.”

4. How about the stability of the material after being used for uranium reduction? More characterization should be provided to show the feasibility of this material for practical application.

Response: We thank the reviewer for the valuable comments. The stability of the material after being used for uranium reduction has been determined by PXRD analysis and the result shows that the used material still maintains its’ initial crystal structure, which proves the stability of the material and the feasibility of the material for practical application. We have added extra statements in the revised manuscript together with additional experimental data in the Supporting Information.

Please refer to **Page 12** in the revised manuscript:

“The PXRD analysis of the used XHOF-TAQ shows that the used material still maintains its’ initial crystal structure, which proves the stability of the material and the feasibility of the material for practical application (**Supplementary Fig. 8**).”

Please refer to **Supplementary Fig. 8** in the revised Supporting Information:

Supplementary Figure 8 PXRD pattern of **XHOF-TAQ** after being used for uranium photoreduction.

5. “The DRS results is consistent with the E_g value obtained from the DFT calculations of the density of electronic states.” How does the E_g value obtained from DFT calculate in the Fig. 4f? Some more references should be cited in this section.

Response: We thank the reviewer for the comment. The lowest unoccupied molecular orbital (LUMO) acts as an electron acceptor, whereas highest occupied molecular orbital (HOMO) represents the ability to donate an electron, the energy gap (E_g) is the possible charge transfer between HOMO and LUMO. In the DOS spectrum of Fig. 4f, the “0” spot represent the LUMO energy and the HOMO is located at 2.43 eV. Therefore, the E_g value can be directly calculated to be 2.43 eV. The corresponding references have been added in the manuscript to support the result.

Please refer to **Reference 59** in the revised manuscript:

“Le Bahers, T., Rerat, M. & Sautet, P. Semiconductors Used in Photovoltaic and Photocatalytic Devices: Assessing Fundamental Properties from DFT. J Phys Chem C 118, 5997-6008 (2014).”

6. The figure in the sentence “...which produce excited electrons to catalyze the reduction of U(VI) to U(IV) (Fig. 1c)” is wrongly written. Fig. 1c should be corrected to Fig. 1d.

Response: We thank the reviewer for the careful revise. The mentioned mistake “Fig. 1c” has been corrected to “Fig. 1d” in the revised manuscript. We have also carefully revised manuscript to correct the existed minor mistakes.

Reviewer: #2

General Comments:

This study reports the fabrication of a novel kind of framework material and the potential application of this material in photoreduction of uranium. This material shows a high uranium recovery capacity and the mechanism for uranium reduction has also been uncovered. This study is an important progress in the area of new crystalline framework materials and uranium immobilization. The finding of this study is interesting to researchers of a broad area. The manuscript is well organized and the result of this study is well presented. The manuscript can be accepted for publication in Nature Communication after addressing the following minor concerns.

General response: We greatly thank the reviewer for taking time to review our manuscript and the positive/valuable comments/suggestions to improve the manuscript. We have revised the manuscript and added some necessary experiments/comparison according to the comments. The revisions of the manuscript have been highlighted by a blue color. We hope our revised manuscript is acceptable.

1. The Cl^- is reported to work as connection nodes in the material fabricated in this study. Can the other halogen ions, such as F^- , Br^- , I^- , can also work as connection nodes for the construction of this kind of material?

Response: We thank the reviewer for the valuable comments. To confirm the ability of the other halogen ions, including F^- , Br^- , and I^- , in the synthesis of this kind of framework material, CuX_2 ($\text{X} = \text{F}$, Br , and I) had also been used to replace CuCl_2 . The result shows that there is no observable crystal product after the replace of CuCl_2 , indicating the other halogen ions cannot take part in the synthesis of such kind of framework material or the reaction condition is not suitable for the synthesis of such kind of framework material containing the other halogen ions. We have added this result in the revised manuscript.

Please refer to **Page 8** in the revised manuscript:

“Furthermore, the replace of CuCl_2 with the other compounds of CuX_2 ($\text{X} = \text{F}$, Br , and I) also cause the failure in synthesis of crystal product, which is because the other

halogen ions cannot take part in the synthesis of such kind of framework material or the reaction condition is not suitable for the synthesis of such kind of framework material containing the other halogen ions.”

2. As the major ligand in the material, TAQ is not stable in air. How about the environmental stability of the material in air?

Response: We thank the reviewer for the valuable comments. The unstable property of TAQ in air is attributing to its' electron rich character, which is easily to be oxidized by the oxygen. After the formation of XHOF, the electron of TAQ is sharing with the Cl^- and the electron rich character of TAQ is reduced, resulting in the increase of the stability of TAQ in XHOF. We have determined the environmental stability of the material in air. The result shows that, after being placed in air for one year, the material still maintains its initial crystal structure, proving the high environmental stability of the material in air. We have added extra statements in the revised manuscript together with additional experimental data in the Supporting Information.

Please refer to **Page 11** in the revised manuscript:

“XHOF-TAQ also shows high environmental stability. After being placed in air for one year, the material still maintains its initial crystal structure (Supplementary Fig. 6).”

Please refer to **Supplementary Fig. 6** in the revised Supporting Information:

Supplementary Figure 6 PXRD pattern of the as-synthesized XHOF-TAQ and the XHOF-TAQ after being placed in air for one year.

3. Cu^{2+} is stated to be essential for synthesis of the material. What's the reason for the essential of Cu^{2+} in the synthesis of this kind of framework material?

Response: We thank the reviewer for the valuable comments. Due to the electron rich characteristic of TAQ, it is easily to be oxidized in air. Thus, the directly use TAQ in synthesis of XHOF-TAQ needs to control the reaction condition strictly, especially the oxygen concentration. According to previous report, cooper can catalyze the transformation of TPAO to form TAQ (Adv. Synth. Catal. **2018**, 360, 334-345). In this study, the CuCl_2 is also used for preparing the material. Thus, the TPAO is transformed into TAQ after the fabrication of the materials. Compared with TAQ, TPAO is stable in air and the use of TPAO doesn't have to control the reaction condition. Thus, the Cu^{2+} is essential in the synthesis of this kind of framework material. The replace of Cu^{2+} with the other metal ions cause the failure in the synthesis of the crystal product, which also proves the essential of Cu^{2+} .

Please refer to **Page 8** in the revised manuscript:

“These results all prove that the TAQ instead of TPAO takes part in the synthesis of XHOF-TAQ. The loss of the hydroxyl group from the oxime group is due to the catalysis of Cu^{2+} during the solvothermal synthesis process.^{48,49} Other compounds of

MCl_n (M = Mn, Zn, Co, Ni, etc.) have also been used to replace CuCl₂ in the synthesis process, but no **observable** crystal product is synthesized, indicating that Cu²⁺ is essential for the synthesis of the crystalline framework material **for its catalytic activity.**”

Please refer to **Reference 49** in the revised manuscript:

“Nandwana, N. K., Dhiman, S., Shelke, G. M. & Kumar, A. Copper-catalyzed tandem Ullmann type C-N coupling and dehydrative cyclization: synthesis of imidazo[1,2-c]quinazolines. *Org. Biomol. Chem.* 14, 1736-1741 (2016).”

4. What’s the purpose to provide the nearest distance between the central Cl⁻ in the manuscript? Dose such characterization associate with the structure stability or the uranium reduction ability of the material?

Response: We thank the reviewer for the valuable comments. The nearest distance between Cl centers is provided to illustrate that the connection type of XHOF is different from the connection of the other framework materials. Generally, halogens can interact with each other, which is descript as X···X, and the distance between them is generally less than 4 Å (*Crystal Growth & Design*, **2005**, 5, 3). In XHOF-TAQ, the distance between Cl atom is 4.36 Å, indicating that there is no interaction between nearest Cl atom, which proves that this material harbors a novel connection type for framework construct. What’s more, without interaction between nearest Cl atom, the Cl connection node can form more hydrogen bonds with the organic ligands and endows the material with high stability. We have added extra statement about the meaning of the distance between Cl atom in the revised manuscript.

Please refer to **Page 10** in the revised manuscript:

“The planes are spatially independent of each other and the nearest distance between the central Cl⁻ of the plane is 4.36 Å, which indicates that there is no interaction between nearest Cl⁻ and, thus, more halogen hydrogen bonds can be formed between Cl⁻ and the organic ligands to provide XHOF-TAQ with higher stability.⁵⁰”

Please refer to **Reference 50** in the revised manuscript:

“Jetti, R. K., Thallapally, P. K., Nangia, A., Lam, C. K. & Mak, T. C. 2,4,6-Tris(4-nitrophenoxy)-1,3,5-triazine: a hexagonal host framework stabilised by the NO₂-trimer supramolecular synthon. *Chem Commun (Camb)*, 952-953 (2002).”

5. The author states that the material fabricated in this study shows comparable thermal stability to other framework materials. Extra references should be provided to support this conclusion.

Response: We thank the reviewer for the valuable advice. According to previous reports, the thermal tolerance of the framework materials is ranging from 200 °C to 600 °C. The thermal tolerance of the material fabricated in this study is 280 °C, which is among this range. We have provided extra reference to support this conclusion.

Please refer to **Reference 52, 53** in the revised manuscript:

“Chui, S. S., Lo, S. M., Charmant, J. P., Orpen, A. G. & Williams, I. D. A chemically functionalizable nanoporous material. *Science* 283, 1148-1150 (1999).

Luo, X. Z. et al. A microporous hydrogen-bonded organic framework: exceptional stability and highly selective adsorption of gas and liquid. *J. Am. Chem. Soc.* 135, 11684-11687 (2013).”

6. How about the photoreduction selectivity of the material? Can it cause the photoreduction of the other metal ions?

Response: We thank the reviewer for the valuable comments. As shown in Fig. 5e, except for uranium, the material also shows weak photoreduction ability to the other metals, including Ni, Co, and Zn. However, attributing to the suitable band gap and band position, XHOF-TAQ shows high selective photoreduction ability to uranium. Comparing with the other metals, the photoreduction ability to uranium is at least 3.68 time higher. We have emphasized the selectivity by providing the comparison of the photoreduction ability of XHOF-TAQ to different metals in the revised manuscript.

Please refer to **Page 16** in the revised manuscript:

“However, the photoreduction ability to uranium is at least 3.68 times higher

than that to the other metals by calculating with the immobilization capacity, proving the high selectivity of XHOF-TAQ in photoreduction of uranium”

7. How about the uranium photoreduction performance of this material comparing with the other available materials? Including the efficiency and the speed of material in uranium reduction.

Response: We thank the reviewer for the valuable comment. The uranium photoreduction performance of this material has been compared with the other available materials by providing extra **Supplementary Table S1**. The result shows that this material is among the best performing material for uranium reduction with high uranium photoreduction capacity together with fastest uranium photoreduction speed. We have added extra statements in the revised manuscript by citing the newly prepared **Supplementary Table S1**.

Please refer to **Page 12** in the revised manuscript:

“Comparing with the other available materials for photoreduction of uranium, XHOF-TAQ exhibits high uranium immobilization capacity together fastest uranium immobilization speed of $34.16 \text{ mg g}^{-1} \text{ min}^{-1}$ under light irradiation (Supplementary Table S1).”

Please refer to **Supplementary Table S1** in the revised Supporting Information:

Table 1 Comparison of uranium photoreduction performance with other available materials.

Catalysts	Dose g L^{-1}	Q_e mg g^{-1}	Efficiency	Equilibrium time	Rate (Q_e/T) $\text{mg g}^{-1} \text{ min}^{-1}$	Ref.
XHOF-TAQ	0.05	1708	85%	50 min	34.16	This work
DI-SNZVI	0.05	427.9	/	3 h	2.38	1
MOF@sponge	0.01	744.6	/	300 min	2.48	2
BP@CNF-MOF	0.01	800	98%	12 h	0.11	3
DC-PAO	0.01	421	/	45 h	0.16	4
AF COF	/	450	99.2%	50 min	9.00	5
$\text{TiO}_2/\text{Fe}_3\text{O}_4$	0.38	252	/	85 min	2.96	6
MOF SCU-19	0.50	500	/	50 h	0.17	7
$\text{Ti}_3\text{C}_2/\text{CdS}$	0.20	242.5	97%	40 min	6.06	8

TiO ₂ /CPAN-AO	0.20	2380	/	5 h	0.79	9
TiO ₂ suspension	2.0	/	98.6%	30 h	/	10
DHBD-TMT	0.125	2640.8	99%	180 min	14.67	11
ZnFe ₂ O ₄	0.20	245	98%	60 min	4.08	12
pTTT-Ben	1.00	4710	78%	4 h	19.63	13
CNBr	0.50	80	95%	20 min	4.00	14
ECUT-SO	0.50	1780	97.8%	60 min	29.67	15
TP-TMT	0.125	2362.4	/	300 min	7.87	16
CdSTe-EDA	0.25	836	97.4%	70 min	11.94	17
CdS/g-C ₃ N ₄	1.00	2,379	99%	20 min	1.14	18
BP-PAO	0.01	1000	80.39 %	32 h	0.52	19
TiO ₂	0.60	3010	98%	300 min	10.03	20
g-CNNs	1.00	/	90%	2 h	/	21
PN-PCN-222	0.50	800	99%	1200 min	0.67	22
g-C ₃ N ₄ /TiO ₂	0.40	25	99%	25 min	1.00	23
MoS ₂ /P-g-C ₃ N ₄	1.00	90	99%	40 min	2.25	24
g-C ₃ N ₄ MCN	0.50	47	99%	20 min	2.35	25
CMPs PTrSO-2	0.50	99.5	99.5%	120 min	0.83	26
CCN-24 g-C ₃ N ₄	0.60	40	100%	50 min	0.80	27
C3N(5)/RGO	0.20	50	94.9%	60 min	0.83	28
Sn-In ₂ S ₃	0.15	/	95%	40 min	/	29
CuS/TNTAs	0.40	115.75	92.6 %	180 min	0.64	30
Te@O-SnS ₂	0.25	704.8	97.3%	60 min	11.75	31
Ti ₃ C ₂ /SrTiO ₃	0.34	115.5	77%	180 min	0.64	32
GA-200	0.40	1050	98%	180 min	5.83	33
PyB-SO ₃ H	0.20	1989	90%	60 min	33.15	34
mGO/g-C ₃ N ₄	0.17	2880.6	96.02%	24 h	2.00	35
MoS ₂ /g-C ₃ N ₄	1.00	33.2	83%	75 min	0.44	36
WO _{2.78}	0.25	507.2	95.6%	120 min	4.23	37
CN550 g-C ₃ N ₄	0.20	1057	99%	1400 min	0.76	38
PFB/CN	0.50	200	99%	100 min	2.00	39
ipCN g-C ₃ N ₄	1.00	/	98%	20 min	/	40
ZnS@/g-C ₃ N ₄	0.20	250	99%	160 min	1.56	41
ZSGCN-5						
rGO KTG	0.10	521.6	92.07%	600 min	0.87	42
ZIF-8/g-C ₃ N ₄	0.10	100	98%	30 min	3.33	43
COF DBD-BTTH	0.01	400	/	6 h	1.11	44
H-VO ₂	0.25	32	95.4%	90 min	0.36	45
Ag/ZIF-8	0.25	433.6	85.8%	20 min	21.68	46
2-PrOH	1.00	59	100%	60 min	0.98	47

BiOBr@COF	0.34	80	91%	540 min	0.15	48
Ag-SnS ₂ @InVO ₄	0.25	120	97.8%	60 min	2.00	49
BCN-80	0.50	800	97.4%	1.5 h	8.89	50
g-C ₃ N ₄ /TiO ₂	0.25	64	80%	250 min	0.26	51
MoS _x /RGO	0.50	15	91.6%	60 min	0.25	52

REVIEWERS' COMMENTS

Reviewer #1 (Remarks to the Author):

The author has carefully revised it as required.

Reviewer #2 (Remarks to the Author):

The authors have provided additional experimental data and adequately responded the comments. The acceptance for publication is suggested.

Title: Halogen hydrogen-bonded organic framework (XHOF) constructed by singlet open-shell diradical for efficient photoreduction of U(VI)

Response to Reviewers' Comments

Reviewer #1:

The author has carefully revised it as required.

General response: We thank the reviewer for taking time to review our manuscript and the positive comments on the manuscript.

Reviewer #2:

The authors have provided additional experimental data and adequately responded the comments. The acceptance for publication is suggested.

General response: We thank the reviewer for taking time to review our manuscript and the positive comments on the manuscript.